# Effect of Insulin Receptor on Juvenile Hormone Signal and Fecundity in *Spodoptera litura* (F.)

**DOI:** 10.3390/insects13080701

**Published:** 2022-08-04

**Authors:** Xue Pan, Yanfang Pei, Cuici Zhang, Yaling Huang, Ling Chen, Liqiong Wei, Chuanren Li, Xiaolin Dong, Xiang Chen

**Affiliations:** 1Hubei Engineering Research Center for Pest Forewarning and Management, College of Agriculture, Yangtze University, Jingzhou 434025, China; 2Key Laboratory of Ecological Environment, Information Atlas (Putian University) Fujian Provincial University, Putian 351103, China

**Keywords:** insulin pathway, juvenile hormone, reproduction, vitellogenin, nutrition

## Abstract

**Simple Summary:**

The tobacco cutworm, *Spodoptera litura* (F.), exemplifies strong reproductive capacities and damages many agricultural crops. The insulin signaling pathway is known as a key determinant of female reproduction in insects. However, the detailed molecular mechanisms in these processes are poorly studied. Here, we injected bovine insulin into the newly emerged moth, resulting in gene expression changes in the insulin pathway, while knockdown of *SlInR* caused an inverse gene expression change involved in the insulin pathway. Further studies indicated that the content of JH-III, Vg, total proteins and triacylgycerol could be suppressed by *SlInR* dsRNA injection. Furthermore, stunted ovaries and lower fecundity were observed by RNAi. Our studies indicated that SlInR plays a key role in JH-III synthesis and the ovarian development in *S. litura*.

**Abstract:**

Insulin signaling can regulate various physiological functions, such as energy metabolism and reproduction and so on, in many insects, including mosquito and locust. However, the molecular mechanism of this physiological process remains elusive. The tobacco cutworm, *Spodoptera litura*, is one of the most important pests of agricultural crops around the world. In this study, phosphoinositide 3-kinase (*SlPI3K*), protein kinase B (*SlAKT*), target of rapamycin (*SlTOR*), ribosomal protein S6 kinase (*SlS6K*) and transcription factor cAMP-response element binding protein (*SlCREB*) genes, except transcription factor forkhead box class O (*SlFoxO*), can be activated by bovine insulin injection. Then, we studied the influence of the insulin receptor gene (*SlInR*) on the reproduction of *S. litura* using RNA interference technology. qRT-PCR analysis revealed that *SlInR* was most abundant in the head. The *SlPI3K*, *SlAKT*, *SlTOR*, *SlS6K* and *SlCREB* genes were decreased, except *SlFoxO,* after the *SlInR* gene knockdown. Further studies revealed that the expression of vitellogenin mRNA and protein, Methoprene-tolerant gene (*SlMet*), could be down-regulated by the injection of dsRNA of *SlInR* significantly. Furthermore, a depletion in the insulin receptor by RNAi significantly decreased the content of juvenile hormone III (JH-III), total proteins and triacylgycerol. These changes indicated that a lack of *SlInR* could impair ovarian development and decrease fecundity in *S. litura*. Our studies contribute to a comprehensive insight into reproduction, regulated by insulin and the juvenile hormone signaling pathway through nutrition, and a provide theoretical basis for the reproduction process in pest insects.

## 1. Introduction

The reproduction of female insects has been studied extensively because of its importance for insect expansion and its potential as a target for pest control [1]. Vitellogenin (Vg), as the precursor of vitellin, acts as the primary source of nutrients for oocyte maturation and ovarian development [2]. Vg is primarily synthesized in insect body fat, then secreted into the haemolymph and utilized by oocytes through a complicated physiological process [3]. The reproductive success of insects depends on Vg biosynthesis and uptake by developing oocytes [4]. Therefore, Vg can be used as a molecular marker of fecundity [5].

The juvenile hormone (JH), a multifunctional insect hormone, is involved in retaining juvenile characteristics in the larval stages and stimulating gonad development in the adult stage in insects [6]. JH exerts its function through its intracellular receptor Methoprene (Met), a member of the family of the basic helix-loop-helix (bHLH)-Per-Arnt-Sim (PAS) transcription factors [7,8]. JH is synthesized by the corpora allata (CA), which are known as another important factor to regulate a large array of physiological processes, such as embryonic development [9], metamorphosis [10], energy homeostasis [11], reproduction [12] behaviors [13,14], etc. JH binds to intracellular receptor Methoprene (Met), a member of the family of the basic helix-loop-helix (bHLH)-Per-Arnt-Sim (PAS) transcription factors, and then the JH signaling pathway is activated [4]. In the adult stage, nutrition is used to stimulate the production of the major gonadotropin JH [15]. As the biosynthesis of Vg is sensitive to the titer of JH [16], many documents have used Vg as a bio-marker to demonstrate the effect on reproduction by JH in insects [17,18,19].

In insects, the insulin signal responses to energy metabolism are well reported. Insulin-like peptides (ILPs) bind to one or more insulin receptors (InRs) [20]. Then, the phosphoinositide 3-kinase (PI3K)-serine/threonine kinase (Akt) signaling pathway or the GTPase Ras/mitogen is activated, in turn [21,22,23,24,25,26]. In these processes, a series of downstream proteins, including the transcription factor (forkhead box class O, FoxO), the target of rapamycin/ribosomal protein S6 kinase (TOR/S6K), transcription factor cAMP-response element binding protein (CREB), etc., are activated by phosphorylation [27,28,29]. The insulin pathway being involved in fecundity of insects is also documented. When female insects reach an adequate nutritional state, the ILPs are secreted into the hemolymph and bound to the InRs. The insulin pathway is activated [30]. Then, the vitellogenin synthesis begins through the control of some hormonal signaling, such as neuropeptides, juvenile hormones (JH) and ecdysteroids [1]. However, most of these studies focus on the upstream regulatory hierarchy. Therefore, it is of great significance to further elucidate the detailed molecular mechanism, especially to identify which genes and hormones participate in cooperation.

## 2. Materials and Methods

### 2.1. Insect Rearing

The *S. liturae* were reared on an artificial diet at 25 ± 1 °C in a 14 L:10 D photoperiod and 70–80% relative humidity (RH) in an incubator (P400G, Ruihua, China) [31]. The newly emerged adults were collected every 12 h and transferred to a one-end open cylindrical plastic cage (Φ = 12 cm, H = 25 cm) covered with filter paper in inner wall for mating and oviposition, and a cotton ball with 10% honey was added as a diet supplement.

### 2.2. Bovine Insulin Microinjection

Following this, 25 mg Bovine insulin (Solarbio, Beijing) was solubilized in 50 μL of 1 M hydrochloric acid and diluted to a final concentration of 2 mg/mL with 0.01 M PBS buffer (pH = 8.0). Each new female adult was injected with 7 μL insulin at the fourth and fifth abdominal junctions after being anesthetized on ice for 1 min using a 10 μL micro-syringe syringe (Hamilton) and the injection point was sealed immediately with white Vaseline (Aladdin, China) [32]. Further, 24 h after injection, three individuals were selected at random from each treated group. Further, the abdomens were collected and pooled for RNA isolation. Females injected with PBS buffer were used as control.

### 2.3. RNA Isolation and Quantitative Real-Time-Polymerase Chain Reaction (qRT-PCR)

Total RNA was isolated from each replicate sample using the Trizol reagent guided by the manufacturer’s suggestion (Invitrogen, Carlsbad, CA, USA). Up to 1 µg of the total was reverse-transcribed into cDNA using M-MLV Reverse Transcriptase (PrimeScript™ II 1st Strand cDNA Synthesis Kit, TaKaRa, Japan). qRT-PCR was performed with a Bio-Rad real-time thermal cycler using the SYBR ExTaq kit (Tli RNaseH Plus) (Takara, Japan) according to the manufacturer’s instructions. Briefly, in a 10 µL total reaction volume containing 1 µL cDNA above, 5 µL TB Green Premix Ex Taq II, 0.4 µL forward and reverse primers (10 µM), respectively, dd water was added to 10 µL. PCR program was as follows: 94 °C for 3 min followed by 40 cycles at 94 °C for 15 s, 58 °C for 30 s and 72 °C for 20 s. The melting curve was analyzed to confirm the amplification of a single fragment. The relative gene expression was analyzed by the 2^−ΔΔCt^ method [33] using (Ribosomal protein L10, RPL10, GenBank Accession No. KC866373) as an internal control. The specific primers for qRT-PCR are shown in Table 1. Quantification of the relative changes in gene transcript level was performed according to the 2^−ΔΔCt^ method. Three biological replicates were performed for all qRT-PCR experiments to obtain the means and standard errors.

### 2.4. Sample Collection for Different Tissue Expression Analysis

To investigate the expression pattern of different tissues, head, thorax, ovaries and fat body from 5 to 8 individuals were carefully dissected and collected. After the tissues were washed in PBS buffer (0.01 M, pH = 8.0) several times, total RNA was isolated from all these samples. Then cDNA was synthesized and gene transcript levels were performed with qRT-PCR as above.

### 2.5. RNA Interference

RNA interference was used to investigate the function of *SlInR* (insulin receptor, GenBank Accession No. XM_022966387.1) with the *EGFP* gene (enhanced green fluorescent protein, GenBank Accession No. DQ768212) as a parallel control. For dsRNA preparation, *SlInR* and *EGFP* genes were first amplified using specific primers conjugated with the T7 RNA polymerase promoter sequence (Table 1). The resultant PCR products were used as templates to synthesize dsRNA (the lengths of ds*SlInR* and ds*EGFP* were 458 (3281–3738 bp) and 547 bp (126–672 bp), respectively) according to the manufacturer recommendations of T7 RiboMAX™ Express RNAi System (Promega, Madison, WI, USA). The quality and integrity of dsRNA were analyzed by agarose gel electrophoresis; meanwhile, the quantity was determined by a Nanodrop1000 Ultramicro spectrophotometer (Implen, München, Germany).

One-day-old virgin adult females were used for RNAi experiment. Before injection, the moths were anesthetized on ice for 5 min. As such, 7 µg dsRNA per individual was injected into the abdomen between the 4th and 5th abdominal segment using a 10 μL micro-syringe (Hamilton) and the injection point was sealed with wax immediately. The injected female was then put into a one-end-open cylinder cage with white paper around the inner wall and two normal males were put together for mating. The cage was 30 cm in height and 15 cm in diameter, the open top was enclosed with a piece of nylon mesh to prevent the insects from moving in or out. Further, 10% honey cotton wool balls were added as nutritional supplement. The gene expression changes were calculated using qRT-PCR at 12 h, 24 h, 36 h and 48 h after dsRNA injection as described above.

For fecundity analysis, the paper in the cage was taken out every day and the number of eggs on the paper was counted for a continuous 6 days.

### 2.6. Determination of Protein and Triglyceride (TAG)

The protein content was determined using a BCA protein concentration assay kit (Beyotime, Shanghai, China). The full abdomen of females was dissected and collected in a 1.5 mL Eppendorf tube. After being weighed, RIPA lysate (250 μL per 20 mg sample) and protease inhibitor (phenylmethylsulfonyl fluoride, PMSF) (final concentration 1 mM) were added. All the mixtures were homogenized on ice for 5 min. Then, 200 μL supernatant was transferred to a 96-well plate and 200 µL BCA developer was added, respectively. After incubating at 37 °C for 30 min, the optical density (OD) values were detected on a HIDEX Sense 425–301 (Turku, Finland) at 562 nm. The standard curve was made according to kit requirements under the same conditions.

TAG assay kit (Solarbio Biotech, Beijing, China) was used for TAG content quantification; the method was described before [34]. In brief, 1 mL n-heptane and isopropanol (1:1) was added to 100 mg dissected ovaries and homogenized for 5 min in an ice bath. After being centrifuged at 8000× *g* for 10 min at 4 °C, 120 μL supernatant was transferred to a new 1.5 mL Eppendorf tube and 375 μL reagent I along with 75 μL reagent II were added in sequence, shocked 30 s on a vortex and stood 10 min at room temperature, 30 μL supernatant was absorbed and transferred to a clean 1.5 mL Eppendorf tube. Then 100 μL reagent III, 30 μL reagent IV, 100 μL reagent V and 100 μL reagent VI were added in sequence. Further, 200 μL mixture was then transferred to a 96-well plate after being incubated for 15 min at 65 °C in an air oven and the OD value was detected on an HIDEX Sense 425-301 (Turku, Finland) at 420 nm. The DD water was used for control to adjust zero.

### 2.7. JH- III Determination

JH- III was purchased from Sigma (St. Louis, MI, USA). All of the other reagents were HPLC grade. The JH- III detection was performed by high-performance liquid chromatography (HPLC) method [35]. Briefly, full females without wings and legs were put in a mortar. After adding 4 mL ether: methanol (V:V = 1:1), homogenized in liquid nitrogen and eluting for 5 min under ultrasonic wave in ice bath, the sample was centrifuged at 8000× *g* for 10 min at 4 °C. The supernatant was transferred to a 15 mL centrifuge tube with 2 mL n-hexane added. After homogenizing in a vortex, the supernatant liquid was extracted and transferred to another new 15 mL centrifuge tube. The extraction was repeated five times. All the supernatant liquids were merged together and the n-hexane was volatilized by vacuum distillation. The residue was re-dissolved using methanol and the final volume was 2 mL. Further, 10 μL sample was separated on a 150 × 2 mm^2^ C18 reverse-phased column (YMC-Pack Pro C18.5 mm, YMC Co., Ltd., Kyoto, Japan) protected by a guard column (YMC-Pack Pro, sphere ODS- H80, YMC Co., Ltd., Kyoto, Japan) with elution of methanol: ddH_2_O (V:V = 70:30) at a flow rate of 0.8 mL/min, using an Agilent 1100 HPLC system with autosampler. The standard curve was made according to gradient concentration of the standard JH-III.

### 2.8. Western Blotting

To analyze the influence of *SlInR* gene silencing on the expression of Vg protein, a DAB kit was used for Western blot, analyzing its expression. The method was modified according to the protocol previously described [36]. Briefly, about 20 mg female fat body was homogenized in 250 µL RIPA lysate. Proteins were separated on a 12% SDS-PAGE gel; the gel was semi-dry transferred for 40 min at 10 volts to an Immobilon-P PVDF membrane (Millipore, Bedford, MA, USA). The immunoreactivity was tested with the anti-SlVg serum (diluted 1:2000) and an IgG goat anti-rabbit antibody conjugated with HRP was used for a secondary antibody (BOSTER, Wuhan, China, 1:5000 dilution) and finally visualized by 3,3- diaminobenzidine carbon tetrachloride (DAB). Non-specific binding was blocked using a 5% fat-free milk solution.

### 2.9. Ovary Dissection and Microscopy

To determine the impact of gene silencing on the development of ovaries in the *S. litura*, the ovaries from the 2nd-day females were separated in 1 × PBS (pH = 8.4) followed by fixation in 3.8% formaldehyde in 1 × PBS for 20 min at room temperature. Dissected ovaries were washed three times for 10 min with 0.2% Triton-X 100 (Sigma, USA) in 1× PBS. After washing, ovaries were photographed with a microscope (Nikon, Sendai, Japan).

### 2.10. Statistical Analysis

The significant differences between the means of multiple samples were analyzed by using one-way analysis of variance (ANOVA), which was followed by the Tukey’s test to perform the multiple comparisons between the means of every two samples. The significant differences between only two samples were determined by using Student’s *t*-test (* *p* < 0.05; ** *p* < 0.01). Statistical analyses were performed using SPSS 20.0 (Chicago, IL, USA) and data are presented as the means ± standard error (SEM).

## 3. Results

### 3.1. Tissue-Specific Expression of SlInR

qRT-PCR was used to characterize the expression pattern of *SlInR* in various tissues in the one-day-old emerging *S. litura* female adults. The results demonstrated that *SlInR* mRNA was mainly expressed in head, thorax and ovaries but weakly in fat body (Figure 1A).

### 3.2. Changes in Insulin Pathway Gene Expression after Bovine Insulin Application

Once insects emerged, female adults were injected with 14 μg bovine insulin or PBS buffer. Gene expression changes in the insulin pathway were quantified by qRT-PCR 24 h later. It showed that the expression levels of *SlPI3K* (GenBank accession No: XM_022968595.1), *SlAKT* (GenBank accession No: XM_022971351.1), *SlTOR* (GenBank accession No: XM_022966802.1), *SlS6K* (GenBank accession No: XM_022972696.1) and *SlCREB* (GenBank accession No: XM_022973491.1) genes were promoted but the *SlFoxO* (GenBank accession No: XM_022969964.1) gene was suppressed significantly (Figure 1B).

### 3.3. Effect of SlInR Silence on Insulin Pathway Gene expression

Newly emerged females were injected with ds*SlInR* or ds*EGFP*. The efficiency of *SlInR* gene knockdown was quantified by qRT-PCR at 12 h, 24 h, 36 h and 48 h. For gene expression changes in the insulin pathway, three individuals were sampled as a pool randomly after 24 h injection and gene expression levels were quantified by qRT-PCR also. The results indicated that the *SlInR* transcripts were decreased 56.4–79.6% after ds*SlInR* injection compared to control (Figure 2A) and the expression levels of *PI3K*, *AKT*, *TOR*, *S6K* and *CREB* genes were significantly decreased, while the FoxO gene was increased 24 h later (Figure 2B).

### 3.4. Effect of SlInR Knockdown on Protein and Triglyceride (TAG) Metabolism

To study whether interference with *SlInR* affects nutrients related to insect reproduction, protein content in female full abdomen and triglyceride titer in ovaries were simultaneously analyzed. In RNAi *SlInR* females, protein and TAG levels were reduced significantly at 12 h, 24 h, 36 h and 48 h (Figure 3A,B). Therefore, the nutritional status in *S. litura* might be regulated by *SlInR* expression at the mRNA level and then modulate reproduction in response to protein and TAG.

### 3.5. Effect of SlInR Silence on SlVg Expression and Reproduction

To detect *SlVg* gene expression in the fat body of *S. litura* after *SlInR* silence, the *SlVg* transcripts were detected at 12 h, 24 h, 36 h and 48 h using the qRT-PCR method. It showed that the *SlVg* expression level was decreased 51.7–78.3% after ds*SlInR* RNA injection vs. ds*EGFP* injection (Figure 4A). For *Sl*Vg protein expression change analysis, three individuals were sampled as a pool randomly after 24 h injection and the protein expression was assayed by Western blotting. The results showed a decrease in Vg protein expression too (Figure 4B). The oviposition of the interfering *SlInR* group was significantly decreased at 6 days compared to the control group (Figure 4C). These results suggest that interference with *SlInR* may affect the eggs laid by impeding oocyte maturation in *S. litura*.

### 3.6. Effect of SlInR Interference on SlMet Expression and JH-III titer

To study whether interference with *SlInR* affects the juvenile hormone (JH) pathway, methoprene-tolerant (*Met*) (GenBank accession No: 2595041) transcripts were detected at 12 h, 24 h, 36 h and 48 h after *SlInR* RNAi. qRT-PCR analysis showed that silencing of *SlInR* greatly decreased the *SlMet* expression in the female full abdomen of ds*InR*-treated adults compared to the controls (Figure 5A). HPLC detection indicated that JH-III titer in the body was reduced dramatically in the ds*SlInR* RNA injection group (Figure 5B). These data indicated that the insulin pathway can affect insect JH cascade.

### 3.7. Effect of SlInR Knockdown on Ovarian Development

Newly emerged insects were injected with ds*SlInR* or ds*EGFP* RNA. Further, 10% honey was added as a nutritional supplement and 48 h later, the ovaries were dissected and photographed. The results suggested that RNAi knockdown of ds*SlInR* resulted in underdeveloped ovaries with abnormal ovarioles in the ovaries (Figure 6).

## 4. Discussion

Insulin binding to the insulin receptor (InR) is usually associated with the regulation of metabolic responses [37]. Previous studies have shown that the InR predominantly stimulated the PI3K/AKT pathway and the Ras/MAPK pathway [38,39]. Our results indicated that bovine insulin could activate the insulin pathway genes (*SlPI3K*, *SlAKT*, *SlTOR*, *SlS6K*, *SlCREB*) (Figure 1B). Bovine insulin is widely used to investigate the physiological functions of ILPs and the related signaling pathway in many insects because the structure and function of ILPs are conservative in different species [21,40,41]. Insect ILPs participate in more physiological processes and act in a different manner compared to mammalian insulins [42]. Each of the ILPs in *A. aegypti* shared some essential residues in the bovine insulin A and B chain [23]. However, the ILPs in *S. litura* have not been reported in the NCBI genome database yet. Therefore, we do not know which ILPs exactly perform this process.

Insulin is known to be a key determinant of female reproduction in insects for regulating ovarian development and oogenesis [23,43,44,45]. In this study, the *SlInR* was effectively knocked down by dsRNA injection (Figure 2A). Contrary to the bovine insulin injection, the expression of critical genes (*SlPI3K*, *SlAKT*, *SlTOR*, *SlS6K*, *SlCREB*) in the insulin signal pathway were decreased after *SlInR* was knocked down (Figure 2B). The expression level of the Vg gene and protein was reduced and, synchronously, fecundity declined after RNAi-mediated knockdown of *SlInR*. Previous studies revealed that knockdown of InR(s) could affect ovarian development in *Culex quinquefasciatus* [46], *Chrysopa pallens* [25], *Bactrocera dorsalis* [44] and so on. Reports also indicated that *AKT*, *TOR*, *PI3K*, *S6K* [47,48], *FoxO* [49] and *CREB* [25] genes are involved in reproduction regulation, respectively, in many insects. Similarly, our results also indicated the important role of the insulin signal pathway in the ovarian development and fecundity of *S. litura*.

Insulin-like peptides (ILPs) bind to one or more InR(s). Then, the phosphoinositide 3-kinase (PI3K)-serine/threonine kinase (Akt) signaling pathway is activated, in turn [21,22]. Further, the downstream genes (*TOR*, *FoxO*, *CREB*, *S6K*) will be activated or have a suppressed response to this process [28,50,51,52]. In this study, our results indicated that the expression levels of genes (*SlPI3K*, *SlAkT*, *SlTOR*, *SlS6K* and *SlCREB*) were increased but the *SlFoxO* gene expression was reduced when the insulin pathway was activated by bovine insulin, while the transcriptions were exactly inverse when the *SlInR* gene was knocked down (Figure 1B and Figure 2B). Similar results were observed in many other insects, such as *Nilaparvata lugens* [51] and *Blattella germanica* [53]. However, a recent report opposed this, as the expression of *FoxO* was reduced when the *InR* gene was knocked down in *B. dorsalis* [44]. Evidence indicated that the *FoxO* gene is a negative regulatory factor in the insulin pathways [40,51,53]. However, it could be activated in different cascades in the insulin pathway [54]. We found the *SlInR* gene was expressed in head, thorax, fat body and female ovaries (Figure 1A). It seems that *SlInR’s* function in different tissues might impact reproduction in a wide variety of ways.

The reproductive success of insects depends on vitellogenin (Vg) biosynthesis and uptake of Vg into developing oocytes mediated by the vitellogenin receptor [4]. Vg synthesis can be induced by JH in many insects [17,19,55]. In this study, the silence of the *SlInR* gene led to a decrease in the *SlMet* gene expression level and decline in JH-III titer (Figure 5A,B). Furthermore, ovarian development was blocked with a reduction in Vg gene transcription and protein translation (Figure 4A,B and Figure 6). These results were presented in *Tribolium castaneum* [40], *Anopheles albimanus* [19] and *Cimex lectularius* [55]. Therefore, how the insulin pathway regulates ovarian development through JH-mediated control of the Vg expression in *S litura* deserves further investigation.

JH has been shown to be involved in the nutrient-mediated regulation of reproduction in many insect species [41,56,57]. The regulation of JH titer by nutrient signals working through the insulin pathway was observed in *Drosophila melanogaster*, *Aedes aegypti* [57], *Blattella germanica* [58] and *Nilaparvata lugens* [59]. Most insect eggs contain large supplies of nutrients to support embryogenesis and oogenesis is commonly a nutrient-limited process [60]. The proteins accumulate in insects developing oocytes [61] and amounts of lipophorins are found in ovaries at different stages of the reproduction cycle in *B. germanica* [62] and *Panstrongylus megistus* [63]. Lipid is also another essential component for the journey to insect reproduction [64,65]. The first nutrition of herbivorous insects is provided by their mothers through the resources they deposit into eggs [66]. Herein, protein concentration in full abdomen and triacylglycerol content in ovaries were significantly decreased after ds*SlInR* RNA injection (Figure 3A,B). Therefore, nutrition is a main factor for limiting reproduction. However, how this reproduction is regulated by insulin pathway crosstalk with the JH signal pathway and nutrition needs to be further elucidated.

## Figures and Tables

**Figure 1 insects-13-00701-f001:**
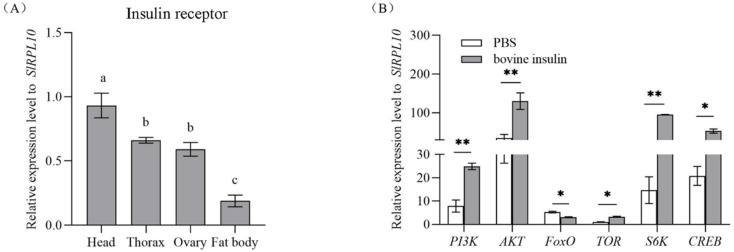
Expression profile of *SlInR* in different tissues and effect of bovine insulin injection on gene expression of insulin signaling pathway. (**A**) qPCR analysis of *SlInR* expression levels in different tissues from one-day-old adult females. (**B**) The expression of insulin signaling pathway genes was detected 24 h after bovine insulin injection by qRT-PCR. Results are represented as means ± SD of three independent samples and samples are normalized to *SlRPL10* expression levels. Different lowercase letters represent significant differences of *SlInR* levels among various tissues determined by one-way ANOVA. Statistically significant difference between the PBS and bovine insulin injection groups (* *p* < 0.05; ** *p* < 0.01, Student’s *t*-test).

**Figure 2 insects-13-00701-f002:**
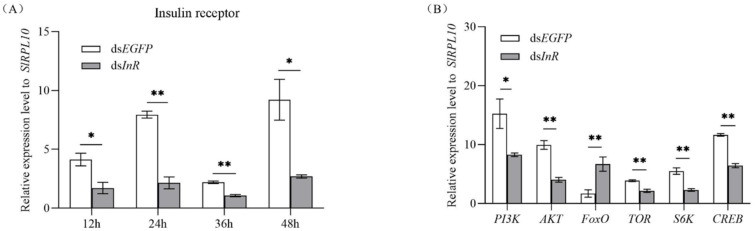
Efficiency of *SlInR* gene knockdown and effect of *SlInR* knockdown on gene expression of insulin signaling pathway. (**A**) Efficiency of *SlInR* gene knockdown at 12, 24, 36 and 48 h. (**B**) The expression of insulin signaling pathway genes was detected 24 h after ds*SlInR* injection by qRT-PCR. Results are represented as means ± SD of three independent samples and samples are normalized to *SlRPL10* expression levels (* *p* < 0.05; ** *p* < 0.01, Student’s *t*-test).

**Figure 3 insects-13-00701-f003:**
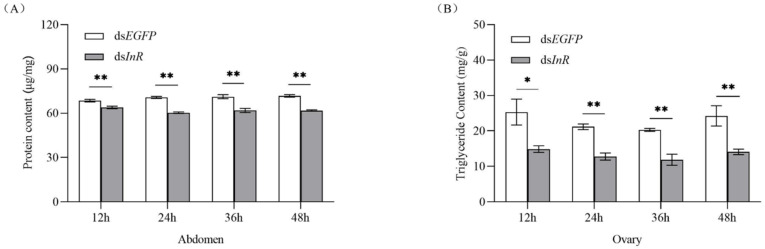
Effect of *SlInR* knockdown on the contents of protein and triglyceride. (**A**) Protein contents in the abdomen after *SlInR* knockdown (*n* = 25). (**B**) Triglyceride contents in ovaries after *SlInR* knockdown (*n* = 25). Statistically significant difference between the ds*EGFP* and ds*InR* injection groups (* *p* < 0.05; ** *p* < 0.01, Student’s *t*-test).

**Figure 4 insects-13-00701-f004:**
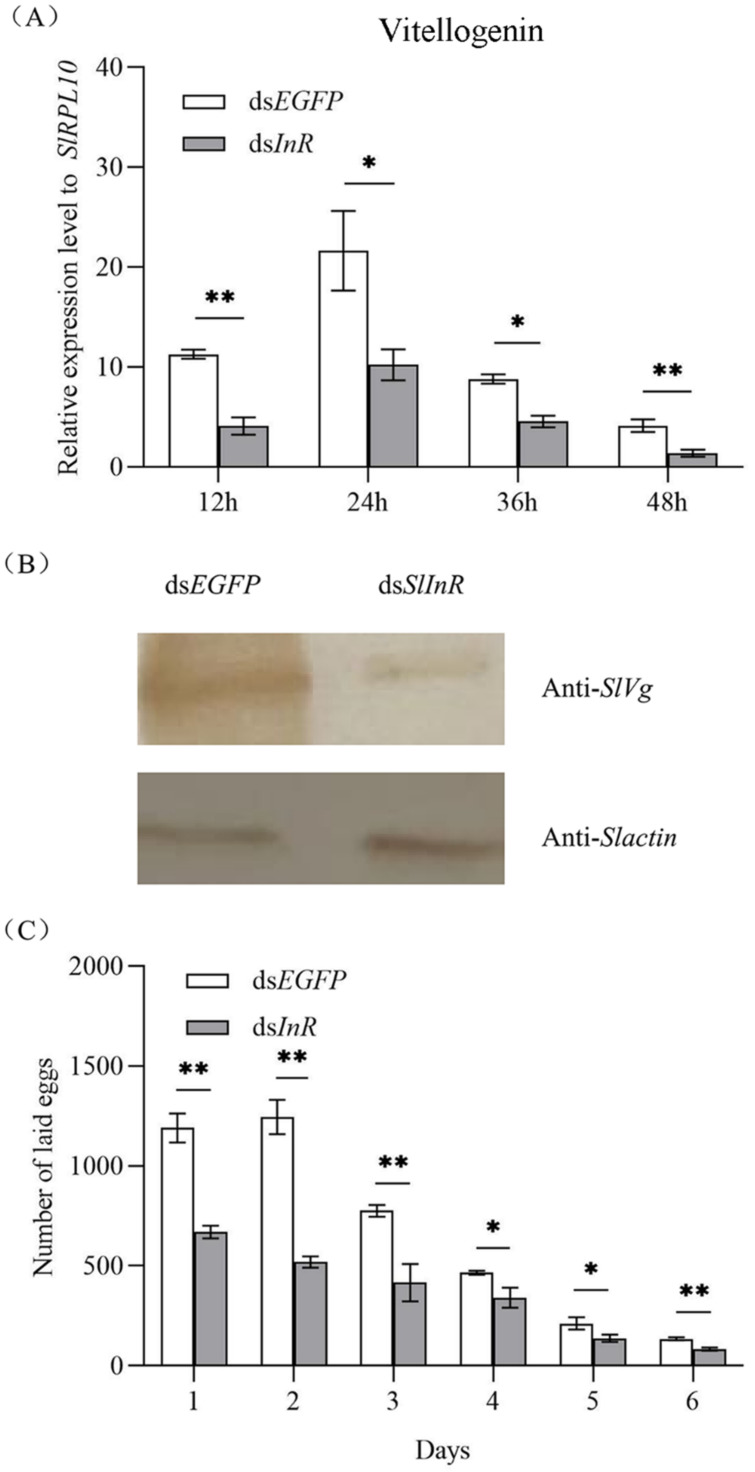
Effect of *SlInR* silence on *SlVg* expression and reproduction. (**A**) qRT-PCR analysis of Vg transcript abundance in abdomen from females treated with ds*SlInR* for 12, 24, 36 and 48 h. (**B**) Western blotting analysis of *SlVg* contents in fat body. Immunoblot of fat body collected 48 h from females injected with ds*SInR* and ds*EGFP* and detection with Vg antibody, an antibody against actin was used as a loading control. (**C**) Daily fecundity. The newly emerged female adults were treated with dsRNA and the fecundity of single female was recorded for six consecutive days after mating (*n* = 25). Statistically significant difference between the ds*EGFP* and ds*InR* injection groups (* *p* < 0.05; ** *p* < 0.01, Student’s *t*-test).

**Figure 5 insects-13-00701-f005:**
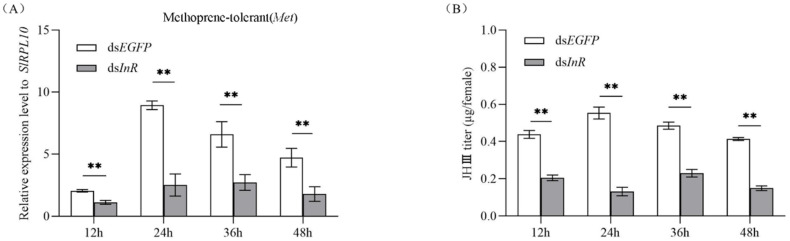
Effect of *SlInR* interference on *SlMet* expression and JH-III titer. (**A**) qRT-PCR analysis of *SlMet* expression levels after *SlInR* knockdown. (**B**) JH-III titer was evaluated after *SlInR* gene knockdown by HPLC. Results are represented as means ± SD of three independent replicates. Statistically significant difference between the ds*EGFP* and ds*InR* injection groups (** *p* < 0.01, Student’s *t*-test).

**Figure 6 insects-13-00701-f006:**
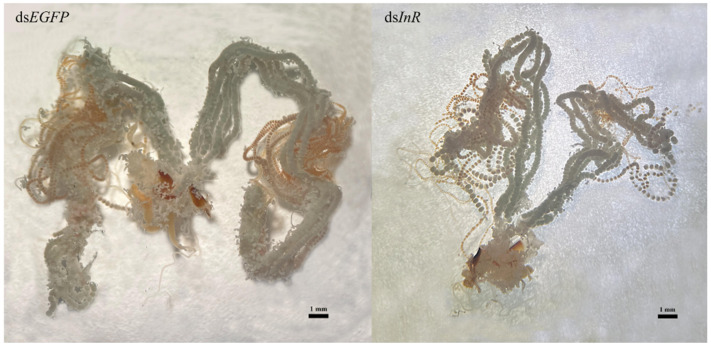
Effect of *SlInR* interference on ovarian development. ds*SlInR* and ds*EGFP* were injected after eclosion and the ovaries of female adults were dissected 48 h later. The scale bar represents 1 mm.

**Table 1 insects-13-00701-t001:** The primer information for qRT-PCR and dsRNA production.

Gene	Name	Sequence (5′-3′)
For qRT-PCR		
*SlInR* (XM_022966387.1)	*SlInR*- qF	CATTGACGCCGTCTGGTTTG
	*SlInR*- qR	CAGCGGAGGTGTTTCCTGAG
*SlPI3K* (XM_022968595.1)	*SlPI3K*- qF	ATACATGACGAGTACGCCCGAG
	*SlPI3K*- qR	TCAGTACGGAGGCAGACAGTAG
*SlAKT* (XM_022971351.1)	*SlAKT*- qF	GGACAAGGACGGACACATCAAG
	*SlAKT*- qR	GCTCAGGATCAGCGAGAACAAC
*SlFoxO* (XM_022969964.1)	*SlFoxO*- qF	AACGGACTTCGGACAAGACG
	*SlFoxO*- qR	TCGCGGACACCCACTATAAG
*SlTOR* (XM_022966802.1)	*SlTOR*- qF	GACAGAACAACAGCCAAGGGAG
	*SlTOR*- qR	CGGAAGTGGAGTCAGAAACAGG
*SlS6K* (XM_022972696.1)	*SlS6K*- qF	AGGACAGACCCAGGATGATGAC
	*SlS6K*- qR	ATAGACGACCCCGAGACTCCAC
*SlCREB* (XM_022973491.1)	*SlCREB*- qF	CTTCAATGACATCCAGGGCGAC
	*SlCREB*- qR	ATTTGATGCTGCTTCCTCCCAC
*SlVg* (EU095334.1)	*SlVg*- qF	CCCACCACACTCTTGCTTTCTC
	*SlVg*- qR	GTCCTTGTTCACGTTCCCTGTC
*SlMet* (2595041)	*SlMet*- qF	ACGCTCTGACAAAGCAAC
	*SlMet*- qR	CTGGCATCGAACTAGACAATAC
*SlRPL10* (KC866373.1)	*SlRPL10*- qF	GACTTGGGTAAGAAGAAG
	*SlRPL10*- qR	GATGACATGGAATGGATG
For *SlInR* dsRNA synthesis		
*SlInR* (XM_022966387.1)	*SlInR* ds- F1	TCGGAATGGTATACGAAGGC
	*SlInR* ds- R1	TCTGGTCATACCGAAGTCCC
	T7 *SlInR* ds- F	GGATCCTAATACGACTCACTATAGGG-TCGGAATGGTATACGAAGGC
	T7 *SlInR* ds- R	GGATCCTAATACGACTCACTATAGGG-TCTGGTCATACCGAAGTCCC
*EGFP* DQ768212	*EGFP* ds- F1	GCTGACCCTGAAGTTCATCTG
	*EGFP* ds- R1	GAACTCCAGCAGGACCATGT
	T7 *EGFP* ds- F	GGATCCTAATACGACTCACTATAGGG-GCTGACCCTGAAGTTCATCTG
	T7 *EGFP* ds- R	GGATCCTAATACGACTCACTATAGGG-GAACTCCAGCAGGACCATGT

## Data Availability

All data is provided in the manuscript.

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
