# Peer review of "Effect of Insulin Receptor on Juvenile Hormone Signal and Fecundity in *Spodoptera litura* (F.)"

_insects, 2022, doi:10.3390/insects13080701_

Round 1
Reviewer 1 Report
Insulin/insulin-like growth factor signaling (IIS) plays various roles in metabolism; and therefore, IIS is also critically important in reproduction. Authors examined relationship between insulin receptor (InR) and fertility and juvenile hormone (JH) signaling that is also important for reproduction by knockdown of InR gene and bovine insulin injection in Spodoptera litura. Although this study may approach important aspects of the research field, the current study is far from publication level mainly because of the following reasons:
1. The presented data are preliminary and superficial. For example, authors analyzed expression changes of factors of IIS after InR knockdown and bovine insulin injection. They also analyzed expression of JH receptor gene and JH titer. However, authors have not studied their interaction. Nothing was revealed how InR affects on JH biosynthesis and JH signaling in this study.
2. The changes in gene expression after treatments were analyzed by qRT-PCR, and higher and lower expression was shown for each gene. The higher expression levels relative to RpL10 of ALL genes are about 1. It is hard to believe; and therefore, the reliability of gene expression data as well as other data is doubtful.
3. There are terrible problems on English. Not only grammar and composition but also structure of paragraphs and logical developments are problematic. It is often hard to read authors’ opinions.
Author Response
Thank you very much for your comments on our manuscript entitled “Effect of insulin receptor on juvenile hormone signal and fecundity in Spodoptera litura (F.)” by Pan et al [Manuscript ID: insects-1750493]. The comments are very helpful for improving our paper. We had made corrections with your comments, and the revised portions are marked in blue script in the text. We hope this revision will be considered for publication in Insects.

Reviewer 2 Report
My comments are marked on the PDF file itself. In general, the article needs a big push for English and grammar before any recommendations can be made. Currently, it's very loose and poorly written. Further please improve the figure quality if possible as currently, it's not up to mark.

Author Response

(The authors gave the same response as above.)

Reviewer 3 Report
The synthesis of vitellogenin and its uptake by maturing oocytes during egg maturation are essential for successful female reproduction. Juvenile hormones act as gonadotropins, regulating vitellogenesis in most insects. Insulin signaling pathway plays an important role in the regulation of insect growth, development and reproduction. The present study inhibits/activates the insulin signaling pathway by InR interference and bovine insulin injection, respectively, indicating that InR is critical for JH biosynthesis and fecundity in Spodoptera litura. Overall, the paper is well written. However, there are still some writing mistakes. It could benefit from some editing to improve agreement of some singular/plural nouns with the conjugation of the corresponding verbs. Below, I provide more detailed comments.
Comments:
1. I think the Introduction is too simple. ‘Many documents have clear demonstrated...’ and ‘Insulin pathway being involved in fecundity of insects in also documented’. Please add more details about these.
2. In Materials and Methods 2.2, There is a lack of reliable references on the homology between bovine insulin and Insect insulin.
3. In table 1, Met family contains two genes, including Met 1 and Met 2. Which gene expression is verified in this article?
4. In results 3.6, why only detect met expression on JH signaling pathway? What about mRNA levels of other genes in the JH signaling pathway?
5. Why 10% honey cotton wool balls used in Materials and Methods 2.5 while 1% honey cotton wool balls used in Materials and Methods 2.1?
6. The insulin signaling pathway is activated after injection of bovine insulin, but what about the changes in ovarian development?
7. The difference significance of the last column (Day 6) in Figure 4C needs further verification.
Author Response
Dear reviewer,
Thank you very much for your comments on our manuscript entitled “Effect of insulin receptor on juvenile hormone signal and fecundity in Spodoptera litura (F.)” by Pan et al [Manuscript ID: insects-1750493]. The comments are very helpful for improving our paper. We had made corrections with your comments, and the revised portions are marked in blue script in the text. We hope this revision will be considered for publication in Insects.

Reviewer 4 Report
It is a paper linking from a mechanistic point of view nutrition and reproduction in Spodoptera litura, specifically describing the regulation mechanisms between insulin, JH and vitellogenin. The proposed signaling pathways are classical and the only objective is to show that they are indeed activated in S. litura. Even if there is no originality in the paper, the description is relatively well done. We are left with the same questions at the end about how insulin acts on the JH pathway. It would have been interesting if the authors had analyzed the biosynthesis and degradation pathway of JH. One also wonders if the effects observed on vitellogenin are dependent on the decrease of JH or if a direct effect of insulin is also to be suspected.
L76 : I am surprised by the volume of 7µL injected. Could you please specify what this volume represents in relation to the estimated volume of the insect. Did you notice a lot of fluid loss at the time of injection?
L78: Can the authors specify the type of wax they use and mention the reference.
L94 : The authors appear to use only one housekeeping gene (RPL10) to relatively analyze the expression of genes of interest. How did they ensure that RPL10 expression did not vary in different tissues or with treatment? The authors should follow the MIQE recommendations more carefully (Bustin et al, 2009)
L127 : Specify if the time kinetics (12h, 24h, 36h, 48h) is performed after RNAi injection or after honey ingestion
L212 : Is SlInsR regulated by insulin?
L228 metablism => “o” is missing
L243: I think the authors meant to mention “SlVg transcripts” and not “SlInR transcripts”
L292 ingestion instead of injestion
L293 a space is missing “pathwaywere”
A summary diagram of the known and assumed pathways of insulin regulation of reproduction would make for easier reading and would be really interesting
Author Response

(The authors gave the same response as above.)

Round 2
Reviewer 1 Report
The revised manuscript has been improved somewhat. Unfortunately, however, it is far from publication level. This reviewer would like to point out following important problems again although there are many other problems.
1. English
The English is still problematic. In addition to multiple grammatical problems, composition of sentences and paragraphs and logical developments are not professional. For some sentences, it is hard to understand even what authors try to say.
2. Relationship between IIS and JH
Authors seem to claim that IIS works mainly on JH production that in turn affects vitellogenesis (p. 2, line 34; p. 15, line 252). Although it is likely according to evidences from other studies with other species, it is not clear from the current study. Authors showed effects of InR knockdown on IIS factors, vitellogenesis, JH titer, and Met expression, these evidences are fragmented and are not linked to each other to explain the biological processes that authors try to describe.
3. Data presentation
qRT-PCR data are stated to be showed using ∆∆Ct method. In ∆∆Ct method, it is necessary to state not only reference gene but also reference sample/condition. In other words, it is unclear what “1” of the relative expression (vertical axes) stands for. It might be better to show relative expression level just to the reference gene but not to reference sample.
The JH III titer is showed as µg per one female. But InR knockdown could affect body size of female. Therefore, it is not suitable to show the data in this way.
Author Response
- English
The English is still problematic. In addition to multiple grammatical problems, composition of sentences and paragraphs and logical developments are not professional. For some sentences, it is hard to understand even what authors try to say.
Thank you for suggestion. Our manuscript had been revised carefully by the authors and re-edited by Elsevier Language Editing Services again.
- Relationship between IIS and JH
Authors seem to claim that IIS works mainly on JH production that in turn affects vitellogenesis (p. 2, line 34; p. 15, line 252). Although it is likely according to evidences from other studies with other species, it is not clear from the current study. Authors showed effects of InR knockdown on IIS factors, vitellogenesis, JH titer, and Met expression, these evidences are fragmented and are not linked to each other to explain the biological processes that authors try to describe.
The aim of this study was designed to investigate details of crosstalk between the insulin pathway with JH biosynthesis and JH signaling which affected the fecundity from the S. litura. We are so sorry the cursory study of the regulation of the insulin pathway affect on fecundity in S. litura. We are now investigating the genes’ function on JH biosynthesis and JH signaling pathway and will provide more evidences in our next research.
- Data presentation
qRT-PCR data are stated to be showed using ∆∆Ct method. In ∆∆Ct method, it is necessary to state not only reference gene but also reference sample/condition. In other words, it is unclear what “1” of the relative expression (vertical axes) stands for. It might be better to show relative expression level just to the reference gene but not to reference sample.
We had revised the figures and correct the vertical axes.
The JH III titer is shown as µg per one female. But InR knockdown could affect body size of female. Therefore, it is not suitable to show the data in this way.
Insulin pathway can affect insect body size had well documented especially on the larval phase. In this study, we injected InR dsRNA after they hab became moths. So we did not observe obvious body size changes after the SlInR gene knockdown within 48h.

Reviewer 2 Report
Dear authors, although there has been some progress made since the previous draft of the document, there are still a great number of errors that need to be addressed. Check out my comments in the accompanying PDF file. Good Luck!

Author Response
Dear authors, although there has been some progress made since the previous draft of the document, there are still a great number of errors that need to be addressed. Check out my comments in the accompanying PDF file. Good Luck!
Thank you for carefully revised our manuscript! The comments are very helpful for improving our paper. We had revised the manuscript carefully according to your suggestion.
Figure 6. The pre-oviposition period of the Spodoptera litura is about 1.8 days on 25℃. So we dissected them after 48 hour. There are many mature eggs in the oviduct (Figure 6 dsEGFP group)

Reviewer 3 Report
Dear authors,
Insulin signaling pathway plays an important role in the regulation of insect growth, development and reproduction. Juvenile hormones act as gonadotropins, regulating vitellogenesis in most insects. The present study inhibits/activates the insulin signaling pathway by InR interference and bovine insulin injection, respectively, indicating that InR is critical for JH biosynthesis and fecundity in Spodoptera litura. Overall, the paper is well written. Through the first revision, most of the written mistakes have been corrected. However, there are still some comments on the results. Below, I provide more detailed comments. I believe this will be a well article after the revision.
Comments:
1. The purpose of this paper is to study the effect of insulin receptor on juvenile hormone signaling and fecundity of S. litura. However, only the expression level of one gene in JH signaling pathway and juvenile hormone titer were used to reflect whether the changes in juvenile hormone signaling pathway were sufficient after InR interference.
2. In this paper, we studied the effect of insulin receptor on the fecundity of S. litura. Then, what are the changes of JH signaling pathway after injection of bovine insulin? What about ovarian development?
Author Response
- The purpose of this paper is to study the effect of insulin receptor on juvenile hormone signaling and fecundity of S. litura. However, only the expression level of one gene in JH signaling pathway and juvenile hormone titer were used to reflect whether the changes in juvenile hormone signaling pathway were sufficient after InR interference.
The aim of this study was designed to investigate details of crosstalk between the insulin pathway with JH biosynthesis and JH signaling which affected the fecundity from the S. litura. We are so sorry the cursory study of the regulation of the insulin pathway affect on fecundity in S. litura. We are now investigating the genes’ function on JH biosynthesis and JH signaling pathway and will provide more evidences in our next research.
- In this paper, we studied the effect of insulin receptor on the fecundity of S. litura. Then, what are the changes of JH signaling pathway after injection of bovine insulin? What about ovarian development?
We are so sorry we ignored this scientific problem in our study. When we injected the bovine insulin into their body, we just considered the genes’ expression changing in insulin pathway and and found ovaries in the end. We will provide these data in our later paper.

Round 3
Reviewer 2 Report
The manuscript got improved during revisions but still has many mistakes. I recommend authors take language expert advice on the manuscript.

Author Response
Comments and Suggestions for Authors:
The manuscript got improved during revisions but still has many mistakes. I recommend authors take language expert advice on the manuscript.
Thank you for your meticulous review of our manuscript! We are so sorry that there are still some mistakes in the manuscript. We had asked the Elsevier Language Editing Services to revised it again with your comments and suggestions.

Round 4
Reviewer 2 Report
Improved!
Author Response
We are so grateful that you are patient to review our manuscript and give us so much suggestion to improve its quality. We had checked the manuscript again and corrected the mistakes.